# Augmenting Genetic Algorithms with Deep Neural Networks for Exploring the Chemical Space

**AkshatKumar Nigam**[1,†]**, Pascal Friederich**[1,2,†]
**Mario Krenn**, [1,3,4]**, Alán Aspuru-Guzik**[1,3,4,5,*]

[1]Department of Computer Science, University of Toronto, Canada.
[2]Institute of Nanotechnology, Karlsruhe Institute of Technology, Germany.
[3]Department of Chemistry, University of Toronto, Canada.
[4]Vector Institute for Artificial Intelligence, Toronto, Canada.
[5]Canadian Institute for Advanced Research (CIFAR) Senior Fellow, Toronto, Canada

## Abstract

Challenges in natural sciences can often be phrased as optimization problems. Machine learning techniques have recently been applied to solve such problems. One example in chemistry is the design of tailor-made organic materials and molecules, which requires efficient methods to explore the chemical space. We present a genetic algorithm (GA) that is enhanced with a neural network (DNN) based discriminator model to improve the diversity of generated molecules and at the same time steer the GA. We show that our algorithm outperforms other generative models in optimization tasks. We furthermore present a way to increase interpretability of genetic algorithms, which helped us to derive design principles.
Our open source implementation: https://github.com/aspuru-guzik-group/GA

## 1 Introduction

The design of optimal structures under constraints is an important problem spanning multiple domains in the physical sciences. Specifically, in chemistry, the design of tailor-made organic materials and molecules requires efficient methods to explore the chemical space. Purely experimental approaches are often time consuming and expensive. Reliable computational tools can accelerate and guide experimental efforts to find new materials faster.

We present a genetic algorithm (GA) (Davis, 1991; Devillers, 1996; Sheridan & Kearsley, 1995; Parrill, 1996) for molecular design that is enhanced with two features:

1. A neural network based adaptive penalty. This promotes exploratory behaviour of the GA and thus improve the diversity of generated molecules.

2. Exploiting the robustness of Selfies (Krenn et al., 2019), we do not need to incorporate any expert based mutation or cross-over rules.

Starting from only simple methane molecules, our algorithm outperforms other generative models in optimization tasks for molecular design. By introducing machine learning (ML) techniques, the long term behaviour is not subject to stagnation, thus solving a significant problem in genetic algorithms (Paszkowicz, 2009).

No domain knowledge is required; thus, our approach is not limited to chemistry-specific questions but can be applied to a wide range of optimization problems.

---

[†]These authors contributed equally
[*]Correspondence to: alan@aspuru.com

## 2 RELATED WORKS

Inverse design is the systematic development of structures with desired properties. In chemistry (Sanchez-Lengeling & Aspuru-Guzik, 2018), the challenge of inverse design has been tackled as an optimization problem, among others in the form of variational autoencoders (VAEs), generative adversarial networks (GANs) and genetic algorithms.

**Variational autoencoders and generative adversarial networks** VAEs (Kingma & Welling, 2013) are a widely used method for direct generation of molecular string or graph representations (Gómez-Bombarelli et al., 2018). They encode discrete representations into a continuous (latent) space. Molecules resembling a known structure can be found by searching around the region of the encoded point. Making using of the continuous latent representation, it is possible to search via gradients or Bayesian Optimization (BO). However, the generation of semantically and syntactically valid molecules is a challenging task. Thus, several follow up works to the VAE have been proposed for inverse design in chemistry. Among them, CVAE (Gómez-Bombarelli et al., 2018), GVAE (Kusner et al., 2017) and SD-VAE (Dai et al., 2018) work directly on string molecular representations. Alternatively, JT-VAE (Jin et al., 2018a) as well as CGVAE Liu et al. (2018) work on molecular graphs. Unlike latent space property optimization, the policy network (PN) based GCPN model (You et al., 2018) proposes a reinforcement learning (RL)-based method for direct optimization on molecular graphs. ORGAN (Guimaraes et al., 2017) demonstrate training string-based generative adversarial networks (GANs) (Goodfellow et al., 2014) via RL. Another method based on adversarial training is VJTNN (Jin et al., 2018b). Segler et al. (2017) first introduced molecule generating models based on language models and reinforcement learning, where actions in an environment are taken to construct a molecule, receiving reward from an external scoring function. This model has also shown strong performance in the GuacaMol benchmark (Brown et al. (2019)).

In all the approaches mentioned above, generative models are trained to mimic the reference data set distributions, thus limiting the exploration ability of VAEs and GANs.

**Genetic algorithms & related methods** There exist several examples of GA based molecule optimization algorithms in literature (O'Boyle et al., 2011; Virshup et al., 2013; Rupakheti et al., 2015; Jensen, 2019; Sheridan & Kearsley, 1995; Parrill, 1996)). While some of these examples pre-define mutations on a SMILES level to ensure validity of the molecules, other approaches use fragment-based assembly of molecules. GAs are likely to get trapped in regions of local optima (Paszkowicz, 2009). Thus, for selection of the best molecules, (Rupakheti et al., 2015; Jensen, 2019) report multiple restarts upon stagnation.

We include the aforementioned models as baselines for numerical comparison.

## 3 GA-D ARCHITECTURE

### 3.1 OVERVIEW

Our approach is illustrated in Figure 1. Our **generator** is a genetic algorithm with a population of molecules $m$. In each generation, the fitness of all molecules is evaluated as a linear combination of molecular **properties** $J(m)$ and the discriminator score $D(m)$:

$$F(m) = J(m) + \beta \cdot D(m) . \tag{1}$$

Random mutations of high **fitness** (best performing) molecules replace inferior members, while best-performing molecules continue to a subsequent generation. The probability of replacing a molecule is evaluated using a smooth logistic function based on a ranking of fitness among the molecules of a generation. At the end of each generation, a neural network based **discriminator** is trained jointly on molecules generated by the GA and a reference **data set**. The fitness evaluation accounts for the discriminator predictions for each molecule. Therefore, the discriminator plays a role in the selection of the subsequent population.

### 3.2 MUTATION & CROSS-OVER RULES

Mutation of molecules to populate subsequent generations is an important element of the GA. A low degree of mutation can lead to a slow exploration of the chemical space, causing stagnation

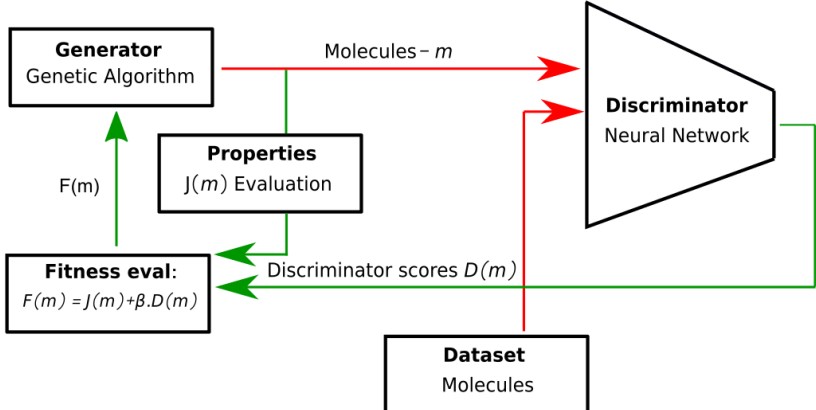

Figure 1: Overview of our hybrid structure, which augments genetic algorithms with ML based neural networks.

of the fitness function. Robustness of SELFIES allows us to do random mutations to molecular strings while preserving their validity. Thus, our mutations only include 50% insertions or 50% replacements of single SELFIES characters. To accelerate the exploration of the GA, we add one domain-specific mutation rule – the direct addition of phenyl groups in approximately 4% of cases. Character deletion is implicitly taken into account in SELFIES mutations, and we do not use cross-over rules.

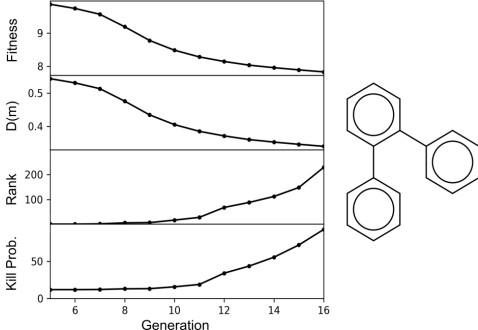

Figure 2: Reducing *Fitness* of *o-terphenyl*, initially possessing the largest fitness in generation 5 (*Rank = 0*). Due to decreasing discriminator predictions $D(m)$, fitness decreases with an increased probability of getting killed (*Kill Prob.*). The molecule is completely replaced in generation 16. *Rank* denotes the position of the molecule in an increasing fitness list of generation molecules.

### 3.3 ROLE OF THE DISCRIMINATOR

The fundamental role of the discriminator is to increase molecular diversity by removing long-surviving molecules. Consider the realistic scenario in which a GA has found a molecule close to a local optimum, where all mutations are lowering the fitness. As a result, this molecule survives for multiple generations while occupying a large fraction of the population. During such periods, the GA has limited view of the chemical space as it repeatedly explores mutations of the same high-fitness molecule.

A straightforward solution would be the addition of a linear penalty (Nanakorn & Meesomklin, 2001) in the fitness function that accounts for the number of successive generations a molecule survives. However, this method assigns independent scores to similar-looking molecules, which again results in less variety.

Our solution is the addition of an adaptive penalty (in our case a neural network based discriminator), thus resolving the problem of stagnation. Molecules with similar representation receive similar classification scores. Furthermore, long-surviving molecules are trained longer and receive weaker scores, resulting in decreasing fitness - reducing the chance of long periods of stagnation (illustrated in Figure 2). The task of the discriminator thus is to memorize families of high performing molecules and penalize their fitness to force the GA to explore different regions in chemical space.

## 4 EXPERIMENTS

Comparing to the literature standard, we aim at maximizing the penalized logP objective $J(m)$ proposed by (Gómez-Bombarelli et al., 2018). For molecule $m$, the penalized logP function is defined as

$$J(m) = \text{logP}(m) - \text{SA}(m) - \text{RingPenalty}(m) , \qquad (2)$$

where logP indicates the water-octanol partition coefficient, SA (Ertl & Schuffenhauer, 2009) represents the synthetic accessibility and prevents the formation of chemically unfeasible molecules, and RingPenalty linearly penalizes the presence of rings of size larger than 6. Our reference data set consists of 250,000 commercially available molecules extracted from the ZINC database (Irwin et al., 2012). All three quantities in the above equation are normalized based on this data set.

Table 1: Comparison of our model with maximum penalized logP scores reported in literature. Models in the upper part are forced to optimize within the distributions of given reference data sets while the GA based approaches in the lower part can freely explore chemical space. Direct comparisons need to take into account these different objectives and scopes. The parameter $\beta$ in our GA can be used to balance exploration and exploitation (see Section 4.6).

| Model | Max. Penalized logP | Model | |
|---|---|---|---|
| GVAE + BO (Kusner et al., 2017) [1] | $2.87 \pm 0.06$ | VAE | |
| SD-VAE (Dai et al., 2018) [1] | $3.50 \pm 0.44$ | VAE | |
| CVAE + BO (Gómez-Bombarelli et al., 2018) [2] | $4.85 \pm 0.17$ | VAE | Exploitation |
| ORGAN (Guimaraes et al., 2017) [1] | $3.52 \pm 0.08$ | GAN | |
| JT-VAE (Jin et al., 2018a) [1] | $4.90 \pm 0.33$ | VAE | |
| ChemTS (Yang et al., 2017) | $5.6 \pm 0.5$ | RNN | |
| GCPN (You et al., 2018) [1] | $7.87 \pm 0.07$ | PN + GAN | |
| Random SELFIES | $6.19 \pm 0.63$ | Random Search | |
| GB-GA (Jensen, 2019) [3] | $7.4 \pm 0.9$ | GA | |
| GB-GA (Jensen, 2019) [4] | $\mathbf{15.76 \pm 5.71}$ | GA | Exploration |
| **GA** (here) | $12.61 \pm 0.81$ | GA | |
| **GA + D** (here) | $13.31 \pm 0.63$ | GA + DNN | |
| (**GA + D(t)** (here) [5] | $\mathbf{20.72 \pm 3.14})$ | GA + DNN | |

[1] average of three best molecules after performing Bayesian optimization on the latent representation of a trained VAE
[2] two best molecules of a single run
[3] averaged over 10 runs with 20 molecules per generation with a molecular weight (excl. hydrogen) smaller than $39.15 \pm 3.50$ g/mol run for 50 generations
[4] averaged over 10 runs with 500 molecules up to 81 characters per generation and 100 generations
[5] averaged over 5 runs with 1000 generations each

### 4.1 UNCONSTRAINED OPTIMIZATION AND COMPARISON WITH OTHER GENERATIVE MODELS

We define our fitness function according to Eq. 1 and 2 with $\beta = 0$ and 10. The algorithm is run for 100 generations with a population size of 500. All generated molecules are constrained to a canonical smile length of 81 characters (as in (Yang et al., 2017; Jensen, 2019)). We train the discriminator (fully connected neural network with ReLU activation and sigmoid output layer, the input is a vector of chemical and geometrical properties characterizing the molecules) at the end of each generation for 10 epochs on 500 molecules proposed by the GA and 500 randomly drawn molecules from the ZINC data set. We report maximum $J(m)$ scores, averaged over 10 independent runs. The highest $J(m)$ achieved by our approach are $13.31 \pm 0.63$ ($\beta = 10$) and

12.61 ± 0.81 ($\beta = 0$), respectively, which is almost twice as high es the highest literature value of 7.87±0.07 (See Table 1). Furthermore, we compare to 50,000 random valid SELFIES strings, which surprisingly outperforms some existing generative models. The GA-D(t) results are explained in the next section.

## 4.2 Long term experiment with a time-dependent adaptive penalty

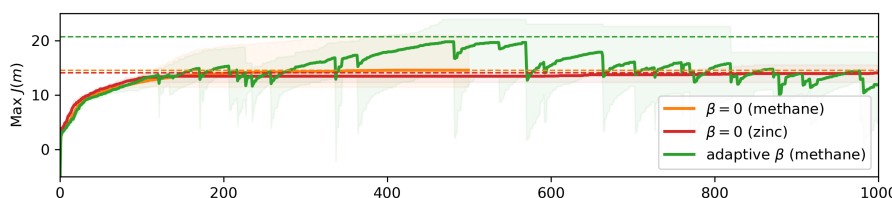

Figure 3: Maximum $J(m)$ values found for 10 independent runs with no discriminator ($\beta = 0$) and 5 independent runs with the introduction of a time-dependent adaptive penalty. The full line is the average of all runs, the shaded areas are the boundaries of all runs and the dashed lines denote the average of all maximal $J(m)$ values found at any generation.

In Figure 3, we show the results of runs where we use a time-dependent adaptive penalty. During periods of saturation, the weight of the discriminator predictions is switched from 0 to 1000 until stagnation is overcome. The genetic algorithm is hence forced to propose new families of molecules to increase $J(m)$. As it can be observed in Figure 3, even after steep decreases in $\max J(m)$, the scores recover and potentially reach values higher than in previous plateaus. We observe that this approach significantly outperforms all previous methods in maximizing the objective $J(m)$.

Visual inspection of the highest performing molecules shows that the GA is exploiting deficiencies of the (penalized) logP metric by generating chemically irrelevant motifs such as sulfur chains. While being of limited relevance for application, this nonetheless shows the that the GA is very efficient in exploring the chemical space and finding solutions for a given task. Analysis of these solutions will help us to better understand and eventually improve objective functions. At the same time, surprising solutions found by an unbiased algorithm can lead to unexpected discovery or boost human creativity (Lehman et al. (2018)). While exploitative tasks that follow some reference database are significant for questions involving drug design, more explorative behaviour could be beneficial in domains such as solar cell design (Yan et al. (2018)), flow battery design (Yang et al. (2018)), and in particular for questions where datas sets are not available, such as design of targets in molecular beam interferometry (Fein et al. (2019)).

## 4.3 Analysis of molecule classes explored by the GA

Figure 4 shows classes of molecules explored by the GA in a trajectory with 1000 generations and a generation size of 500 (see trajectories in Figure 3). A K-means clustering analysis based in the RDKit fingerprint bit-vectors with 20 clusters was used to automatically generate labels for the 50 best performing molecules (in terms of their properties $J(m)$) in each generation. We find that the algorithm starts with a class of relatively small molecules ($\approx 40$ generations, see class 10), after which the algorithm explores several different classes of large molecules with *e.g.* aromatic rings and conjugates carbon chains (see class 15) and long linear sulfur chains (see class 1). Class 1 includes the molecules with the highest $J(m)$ scores found in this work, exceeding values of 20.

The clustering analysis furthermore helps to analyze the large number of generated molecules. This allows us to learn from the data and derive design rules to achieve high target properties. In case of the penalized logP score, we find that representative examples (short distance to the cluster center) of high performing classes contain aromatic rings, conjugated carbon chains and linear sulfur chains, which can act as design rules for molecules with high penalized logP scores. Using these design rules to construct molecules of the maximally allowed length consisting of only sulfur chains, conjugated carbon chains and chains of benzene rings with sulfur bridges yields $J(m)$ scores of 31.79, 8.58 and 10.02, respectively. The $J(m)$ score reached by the linear sulfur chain outperforms the best scores found by the GA and other generative models.

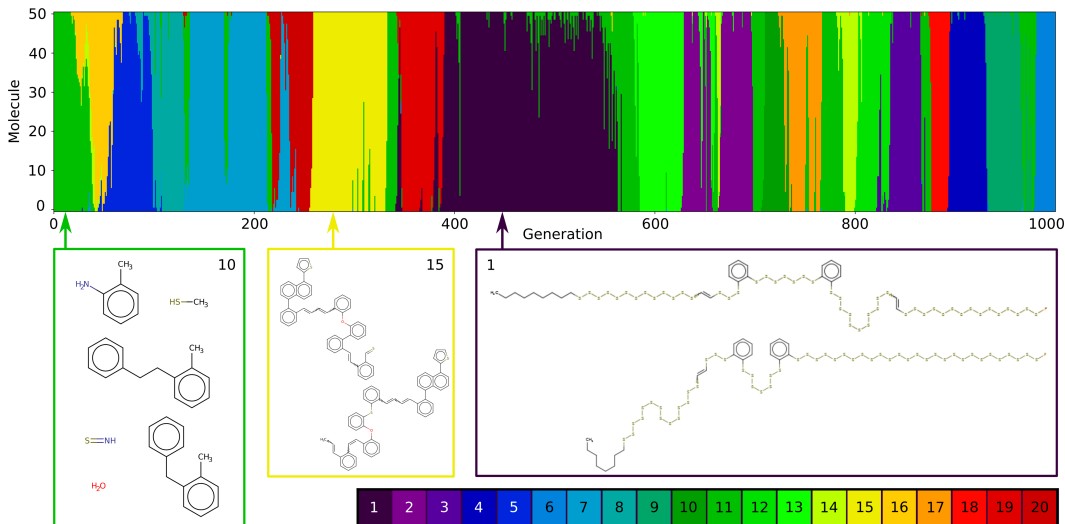

Figure 4: Classes of molecules explored by the GA.

To visualize the explorative behaviour of the GA, we did a two-dimensional principal component analysis (PCA) of the molecular fingerprint vectors of all molecules generated by the GA-D(t) in the trajectory shown in Figure 4. Five snapshots of the trajectory are shown in Figure 5, colored according to their chemical family. The property score $J(m)$ of the trajectory is visualized in the images. We find that the GA sequentially explores different parts of the 2D projection of the chemical space, finding different families of high performing molecules, while the discriminator prevents the algorithm from searching in the same space repeatedly.

## 4.4 CONSTRAINED OPTIMIZATION

In the previous sections, we aimed to maximize the penalized logP objective. Here, we consider two different tasks: firstly, generating molecules of specific chemical interest and secondly, modifying existing molecules to increase penalized logP scores. We used $\beta = 0$ throughout Section 4.4.

### 4.4.1 GENERATING MOLECULES WITH SPECIFIC PROPERTIES

We run 250 instances of randomly selected sets of target properties (logP in the interval from -5 to 10, SA scores in the range from 1 to 5 and ring penalty for 0 to 3). The fitness function is modified to minimize the summed squared difference between the actual and desired properties. We compute the number of experiments that successfully proposed molecules with a squared difference less than 1.0. Each run is constrained to run for 100 generations with a maximum canonical SMILES length of 81 characters. In 90.0% of the cases, our approach proposes the desired molecules.

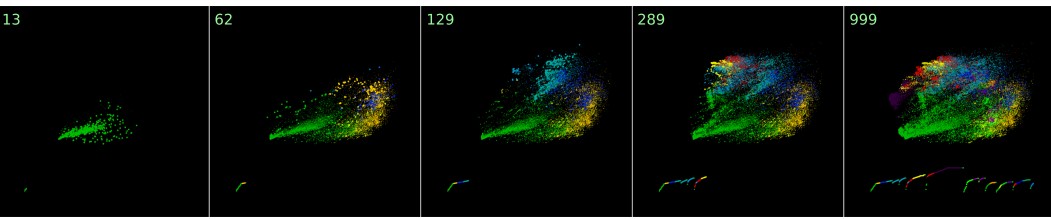

Figure 5: Two-dimensional PCA of five time snapshots of the molecular space explored by the GA-D(t).

Table 2: Comparison on constrained improvement of penalized logP of specific molecules.

| | δ=0.4 | | δ=0.6 | |
| --- | --- | --- | --- | --- |
| | **Improvement** | **Success** | **Improvement** | **Success** |
| JT-VAE (Jin et al., 2018a) | $0.84 \pm 1.45$ | 83.6% | $0.21 \pm 0.71$ | 46.4% |
| GCPN (You et al., 2018) | $2.49 \pm 1.30$ | **100.0%** | $0.79 \pm 0.63$ | **100.0%** |
| MMPA (Jin et al., 2018b) | $3.29 \pm 1.12$ | - | $1.65 \pm 1.44$ | - |
| DEFactor (Assouel et al., 2018) | $3.41 \pm 1.67$ | 85.9% | $1.55 \pm 1.19$ | 72.6% |
| VJTNN (Jin et al., 2018b) | $3.55 \pm 1.67$ | - | $2.33 \pm 1.17$ | - |
| **GA** (here) | $\mathbf{5.93 \pm 1.41}$ | **100.0%** | $\mathbf{3.44 \pm 1.09}$ | 99.8% |

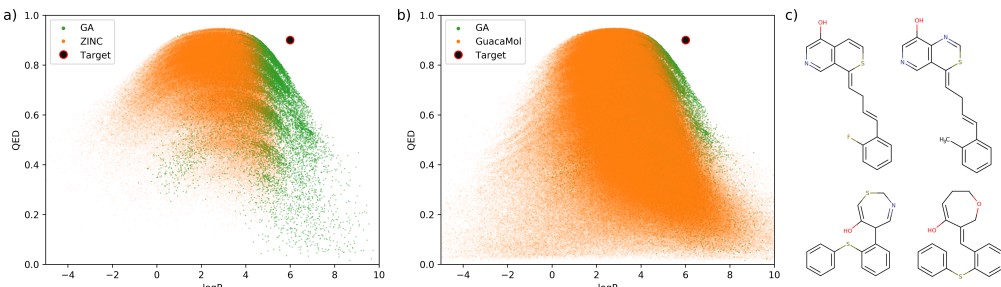

Figure 6: Distributions of logP and QED for a) ZINC and b) GuacaMol data set compared to molecules generated using the GA. c) Examples of molecules generated by the GA with high logP and QED scores.

### 4.4.2 IMPROVING PENALIZED LOGP SCORES OF SPECIFIC MOLECULES

In this experiment, we follow the experimental setup proposed by You et al. (2018). We optimize the penalized logP score of 800 low-scoring molecules from the ZINC data set. Our genetic algorithm is initiated with a molecule from the data set, and we run each experiment for 20 generations and a population size of 500 without the discriminator. For each run, we report the molecule $m$ that increases the penalized logP the greatest, while possessing a similarity $sim(m, m') > \delta$ with the respective reference molecules $m'$. We calculate molecular similarity based on Morgan Fingerprints of radius 2. To ensure generation of molecules possessing a certain similarity, for molecule $m$ we modify the fitness to:

$$F(m) = J(m) + \text{SimilarityPenalty}(m) . \tag{3}$$

Here, SimilarityPenalty$(m)$ is 0 if $sim(m, m') > \delta$ and $-10^6$ otherwise. In Table 2, we report the average improvement for each molecule. Success is determined when the GA successfully improve upon the penalized logP score, while not violating the similarity constraint. Figure S1 shows examples of improved molecular structures.

### 4.5 SIMULTANEOUS LOGP AND QED OPTIMIZATION

To show the performance of the GA on a drug-discovery task (Brown et al., 2019; Polykovskiy et al., 2018), we modified the objective function to include the drug-likeness score QED (Bickerton et al., 2012). Solubility metric logP and drug-likeness cannot be maximized at the same time, which is shown in Figure 6 at the example of the ZINC and the GuacaMol data set (Brown et al., 2019). The experiment shows that the GA is able to efficiently and densely sample the edge of the property distributions of both data sets. Example molecules that simultaneously maximize logP and QED are shown in Figure 6c.

### 4.6 MODIFICATION OF THE HYPERPARAMETER $\beta$

The definition of the fitness function used in this work (see Equation 1) has a free parameter ($\beta$), that balances the relative importance of molecular target properties $J(m)$ and discriminator score

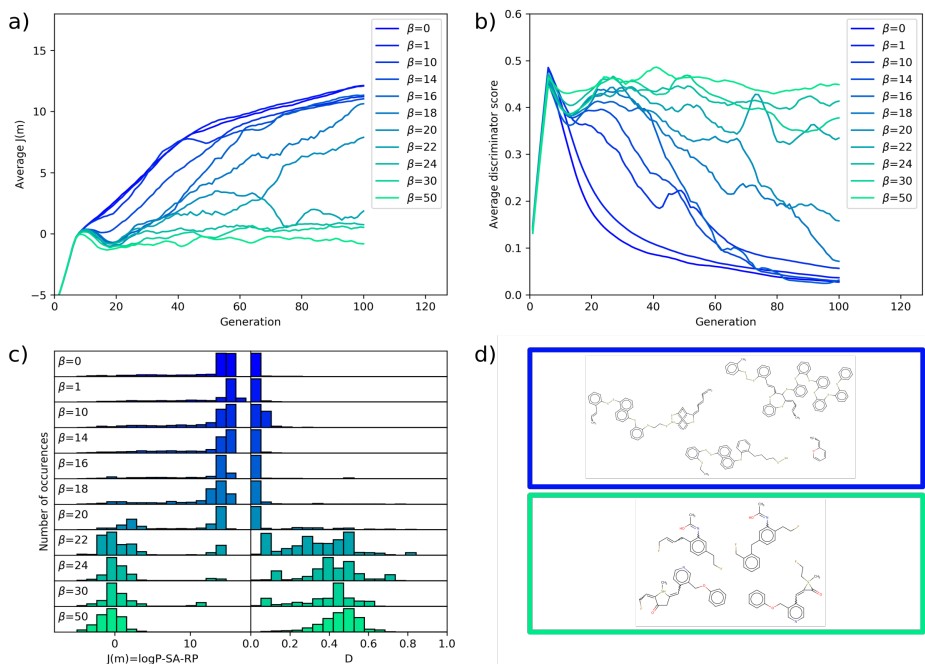

Figure 7: Variation of the model parameter $\beta$ that weights the discriminator score in the overall fitness function. a) Average $J(m)$ as a function of the generation. b) Average discriminator score as a function of the generation. c) Distributions of $J(m)$ and discriminator score $D$ for multiple values of $\beta$. d) Examples of molecules generated with $\beta = 0$ and $\beta = 50$.

$D(m)$. Large values of $\beta$ promote the generation of molecules that resemble the molecules from the reference data set, while small values of $\beta$ let the GA explore molecules outside of the distribution of the reference data set. To systematically explore this behaviour, we varied the $\beta$ parameter and analyzed the resulting values of $J(m)$, $D(m)$ as well as their distributions. Figure 7 illustrates the results of this test, where each curve is obtained from averaging five independent runs with equal settings.

In Figure 7a, we show the average property score $J(m)$ for 11 different values $\beta$ and observe, as expected, that low values of $\beta$ on average lead to higher $J(m)$ score while very high values of $\beta$ limit the algorithm to stay inside the reference data set distribution, which, per definition, has a mean $J(m)$ score of 0. Figure 7b illustrates the low average discriminator scores encountered after a few generations in case of low values of $\beta$. The curves shown in Figure 7a and b suggest that $\beta$ allows us to interpolate between high and low $J(m)$ and high and low resemblance with the reference data set. However, as shown in Figure 7c, this is not the case. At values of $\beta \approx 20$, there is a rather fast transition from molecules with high values of $J(m)$ of approximately 12-13 to a $J(m)$ distribution centered around 0 which is similar to the $J(m)$ distribution of the reference data set. The same abrupt change is visible in the distributions of the discriminator scores. Only at $\beta \approx 18$, we find a slightly wider distribution of intermediate $J(m)$ values, which we assume is partially related to the fact that the $\beta = 18$ curve shown in Figure 7a is not converged yet.

Figure 7d shows examples of molecules generated with $\beta = 0$ (upper panel) and $\beta = 50$ (lower panel). While in the first case, the GA finds large molecules with many aromatic rings and long linear chains, that do not resemble the molecules in the reference data set, the latter case shows molecules with structures and properties resembling the reference data set. In this case, the distribution of $J(m)$ is comparable to that of the data set (with 0 mean and unit standard deviation), while the discriminator encounters difficulty in correctly classification of the GA molecules, indicated by a mean discriminator score varying around 0.5 across many generations.

## 5 CONCLUSIONS AND FUTURE WORK

We presented a hybrid GA and ML-based generative model and demonstrated its application in molecular design. The model outperforms literature approaches in generating molecules with desired properties. A detailed analysis of the data generated by the genetic algorithm allowed us to

interpret the model and learn rules for the design of high performing molecules. This human expert design inspired from GA molecules outperformed all molecules created by generative models.

For computationally more expensive property evaluations, we will extend our approach by the introduction of an on-the-fly trained ML property evaluation method, which will open new ways of solving the inverse design challenge in chemistry and materials sciences.

Our approach is independent of domain knowledge, thus applicable to design questions in other scientific disciplines beyond chemistry. We therefore plan to generalize the GA-D approach to make it a more general concept of generative modelling.

ACKNOWLEDGMENTS

The authors thank Gabriel dos Passos Gomes & Andrés Aguilar Granda for useful discussions. A. A.-G. acknowledges generous support from the Canada 150 Research Chair Program, Tata Steel, Anders G. Froseth, and the Office of Naval Research. M.K. acknowledges support from the Austrian Science Fund (FWF) through the Erwin Schrödinger fellowship No. J4309. P.F. has received funding from the European Union's Horizon 2020 research and innovation programme under the Marie Sklodowska-Curie grant agreement No 795206.

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

## 6 SUPPLEMENTARY INFORMATION

Figure S1 shows examples of the molecules optimized in Section 4.4.

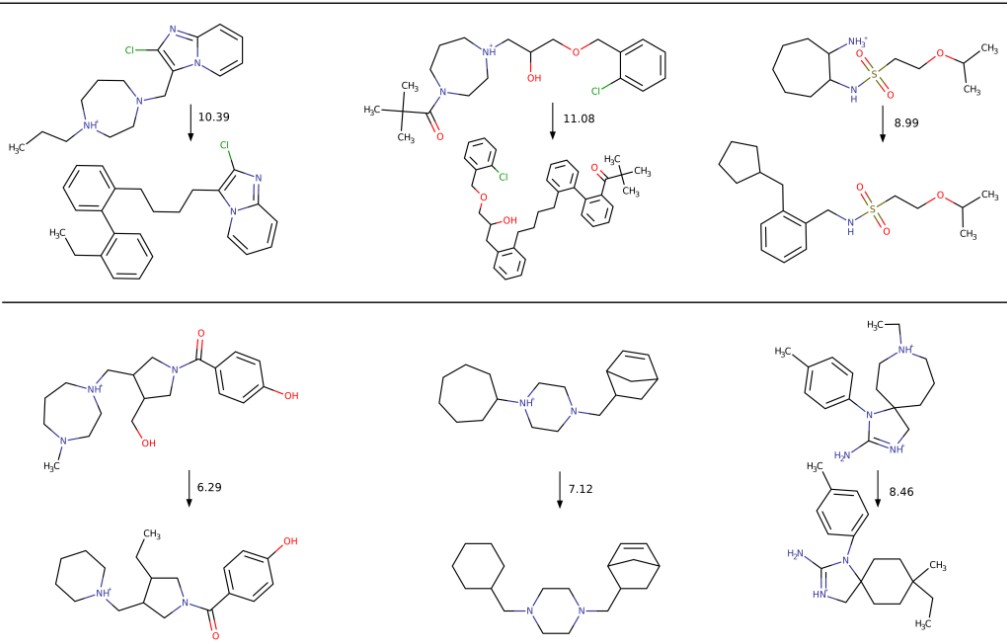

Figure S1: Molecular modifications resulting in increased penalized logP scores under similarity constraint $sim(m, m') > 0.4, 0.6$. We show the molecules that resulted in largest score improvement.

Figures S2-S4 show comparisons between the property distributions observed in molecule data sets such as the ZINC and the GuacaMol data set and property distributions of molecules generated using random SELFIES (Figure S2), GA generated molecules with the penalized logP objective (Figure S3) and GA generated molecules with an objective function which includes logP and QED (Figure S4). While the average logP scores of average SELFIES are low, the tail of the distribution reaches to high values, explaining the surprisingly high penalized logP scores shown in Table 1. The QED and weight distributions of molecules optimized using the penalized logP objective significantly differ from the distributions of the ZINC and the GuacaMol data set (see Figure S3). As soon as the QED score is simultaneously optimized, the distributions of GA generated molecules and molecules from the reference data sets become more similar (see Figure S4). Figure 6 shows that the GA can simultaneously optimize logP and QED.

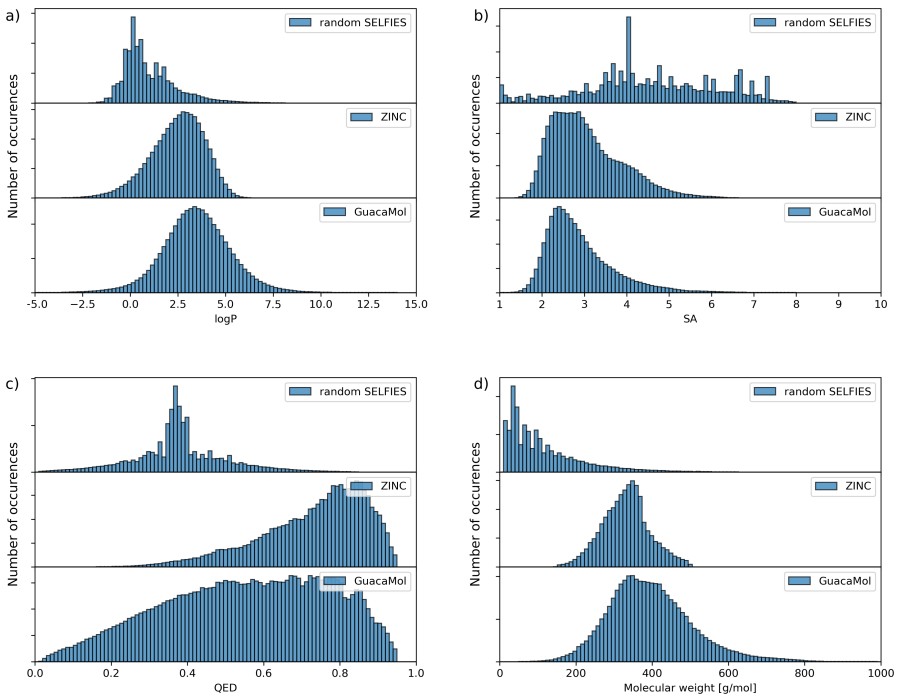

Figure S2: Distributions of a) logP, b) SA, c) QED and d) molecular weight for randomly generated SELFIES, molecules from the ZINC data set and molecules from the GuacaMol data set.

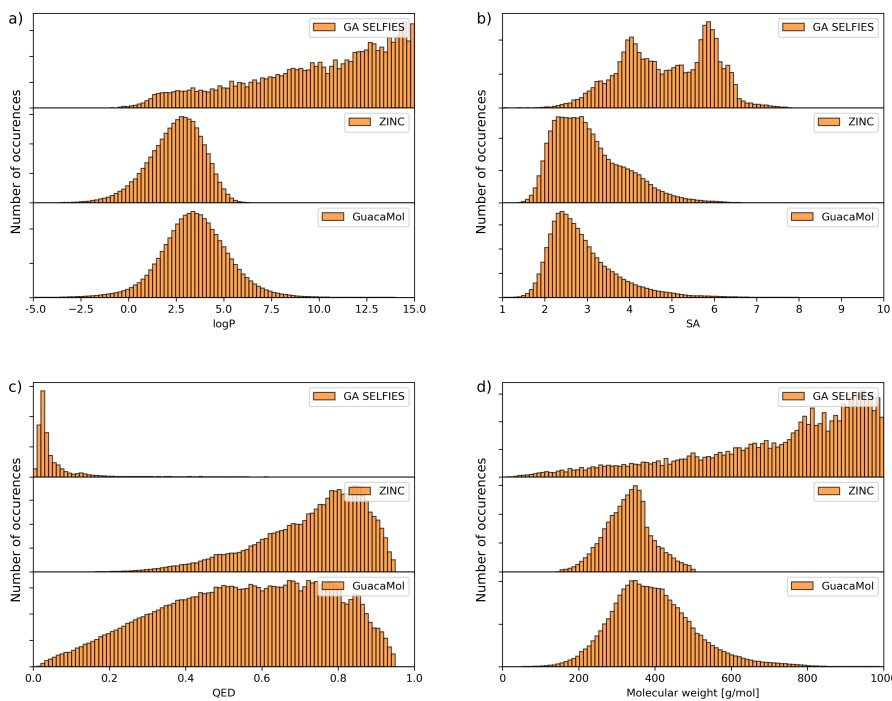

Figure S3: Distributions of a) logP, b) SA, c) QED and d) molecular weight for GA generated SELFIES (penalized logP objective function), molecules from the ZINC data set and molecules from the GuacaMol data set.

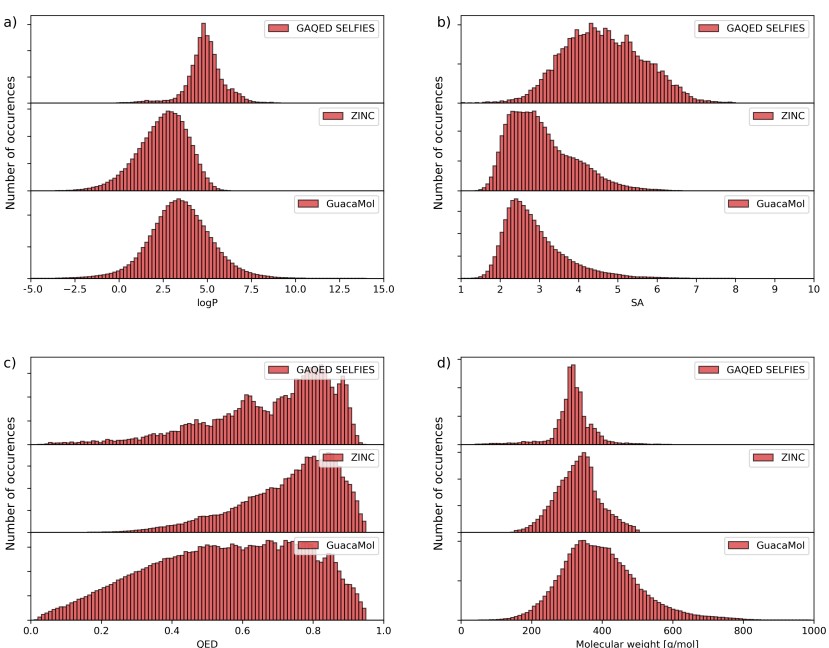

Figure S4: Distributions of a) logP, b) SA, c) QED and d) molecular weight for GA generated SELFIES (logP and QED as objective function), molecules from the ZINC data set and molecules from the GuacaMol data set.

