# OpenReview forum: "Augmenting Genetic Algorithms with Deep Neural Networks for Exploring the Chemical Space"
_ICLR.cc/2020/Conference — Accept (Poster)_

### Official Review · AnonReviewer1 · 2019-10-24
**Official Blind Review #1**

**Rating:** 3

**Review:**

Generating novel drug molecules is an important problem. Prior approaches to solve this problem has explored approaches like VAEs, GANs and Genetic Algorithms. The authors propose a genetic algorithm that is augmented with Neural Networks for this purpose. The authors claim impressive performance with their algorithms. While maximizing for Penalized logP, the molecules that they generate have almost twice the maximum Penalized logP reported in the literature.

From a purely molecular optimization point of view, their results are impressive. However, I suspect the paper does not clearly illuminate what is going on. For example, the Random SELFIES performs twice as good as state of the art generative models from couple of years back like JT-VAE, SD-VAE and Grammar VAE.  I suspect what is happening is that the random model and the models proposed by the authors are veering too much from the training data. The authors indeed suggest that this is the case in the discussion about the parameter beta. For small values of beta, the generated molecules move away from the reference distribution. While standard VAE models tries to ensure that the training and generated models are from similar distributions, the models proposed by the authors face no such constraint, for small values of beta. This makes the comparison with other models unfair - because other models try their best to generate molecules from the reference distribution, while at the same time trying to optimize for molecular properties like higher penalized logP. If those models had the freedom not to restrict its exploration to the training distribution, one can obviously generate molecules with higher penalized logP.

The authors seem to suggest that moving away from the reference distribution might be desirable, if we can get molecules with higher penalized logP. Indeed, this is not the case. Zinc is a database of small drug-like molecules. When they move away from this distribution, they will start generating molecules which are not drug-like and which perform poorly in other metrics like QED and Synthetic Accessibility. In this sense, the results claimed by the authors are misleading.

Can the authors plot a distribution of molecular properties (molecular weight, Synthetic Accessibility, QED, logP and number of Rings) for zinc, and the generated molecules with different values of beta, so that we can analyze this phenomenon better? Code for this (along with examples) can be found in the MOSES toolkit: https://github.com/molecularsets/moses

**Experience Assessment:**

I have published one or two papers in this area.

**Review Assessment: Checking Correctness Of Derivations And Theory:**

N/A

**Review Assessment: Checking Correctness Of Experiments:**

I carefully checked the experiments.

**Review Assessment: Thoroughness In Paper Reading:**

I read the paper thoroughly.

---

> ### Author Response · Authors · 2019-11-15
> **Reply 1**
>
> Reviewer comment: Generating novel drug molecules is an important problem. Prior approaches to solve this problem has explored approaches like VAEs, GANs and Genetic Algorithms. The authors propose a genetic algorithm that is augmented with Neural Networks for this purpose. The authors claim impressive performance with their algorithms. While maximizing for Penalized logP, the molecules that they generate have almost twice the maximum Penalized logP reported in the literature.
>
> From a purely molecular optimization point of view, their results are impressive. However, I suspect the paper does not clearly illuminate what is going on. For example, the Random SELFIES performs twice as good as state of the art generative models from couple of years back like JT-VAE, SD-VAE and Grammar VAE. I suspect what is happening is that the random model and the models proposed by the authors are veering too much from the training data. The authors indeed suggest that this is the case in the discussion about the parameter beta. For small values of beta, the generated molecules move away from the reference distribution. While standard VAE models tries to ensure that the training and generated models are from similar distributions, the models proposed by the authors face no such constraint, for small values of beta. This makes the comparison with other models unfair - because other models try their best to generate molecules from the reference distribution, while at the same time trying to optimize for molecular properties like higher penalized logP. If those models had the freedom not to restrict its exploration to the training distribution, one can obviously generate molecules with higher penalized logP.
>
> Answer: We thank the reviewer for his useful and constructive comments and suggestions. We agree that there is a significant difference between algorithms such as VAEs and GANs that are intrinsically forced to stay within the distribution of the respective reference data set and genetic algorithms which are freely exploring the parameter space (in our case molecular space). To make this distinction immediately clear, we modified Table 1 and separated the models that are bound to stay within the reference distribution from the explorative genetic algorithms (the new caption includes: “Models in the upper part are forced to optimize within the distributions of given reference data sets while the GA based approaches in the lower part can freely explore chemical space. Direct comparisons need to take into account these different objectives and scopes. he parameter beta in our GA can be used to balance exploration and exploitation (see Section 4.6)”). We agree that there are tasks where staying within a reference distribution is useful and important, but we want to emphasize that algorithms that efficiently find new good solutions outside of what is currently known (und thus part of the reference data set) are also very desirable for molecular design tasks. Finding new good solutions (or even just finding deficiencies of the objective function) will help human researchers to build a better intuition of the task and thus enable a faster improvement and potentially the discovery of completely new solutions, allowing for the inspiration of new creative and surprising ideas.

---

> ### Author Response · Authors · 2019-11-15
> **Reply 2**
>
> Reviewer comment: The authors seem to suggest that moving away from the reference distribution might be desirable, if we can get molecules with higher penalized logP. Indeed, this is not the case. Zinc is a database of small drug-like molecules. When they move away from this distribution, they will start generating molecules which are not drug-like and which perform poorly in other metrics like QED and Synthetic Accessibility. In this sense, the results claimed by the authors are misleading.
>
> Answer: We agree with the reviewer that for certain tasks, including the important task of drug discovery, a real-world objective function differs significantly from the penalized logP objective that is used to test many generative models. However, we should differentiate between the objective function that was optimized and the model that we propose to optimize arbitrary objective functions. We therefore performed additional experiments with a modified objective function that includes the QED score, on top of the previously used penalized logP metric (which includes logP and SA), see Fig. 6. Apart from that, there is a wide range of applications of generative models outside of drug-discovery, including organic electronics (e.g. for clean energy applications such as organic solar cells or optoelectronic applications such as organic light emitting diodes) as well as for energy storage applications such as organic flow batteries, tasks in quantum interferometry and spectroscopy, and many others. In many aspects, the objectives as well as the available reference data sets significantly differ from drug-discovery tasks and an explorative behaviour is absolutely required. One example is the search for alternatives to non-fullerene acceptor materials for organic solar cells. There are multiple examples of molecules that achieve high power conversion efficiencies (which is the main objective for solar cells). However, these molecules are far too complex to be produced on an industrial scale, which makes is mandatory to explore the chemical space outside of their distribution to find molecules with similar performance but significantly lower complexity and higher synthesizability.
>
> Reviewer comment: Can the authors plot a distribution of molecular properties (molecular weight, Synthetic Accessibility, QED, logP and number of Rings) for zinc, and the generated molecules with different values of beta, so that we can analyze this phenomenon better? Code for this (along with examples) can be found in the MOSES toolkit: https://github.com/molecularsets/moses
>
> Answer: We added the requested plots to the SI. They show that the random SELFIES as well as GA generated molecules that were found when optimizing logP and SA perform poorly on the QED metric (as expected) and have a different weight distribution as the ZINC and the GuacaMol data sets. However, when simultaneously optimizing logP and QED, this situation changes, showing that a proper choice of the objective function leads to desired results (see also reply to Reviewer 2). We therefore performed a new experiment (see also new Section 4.5 and Figure 6) on simultaneous optimization of logP and QED with a comparison to the ZINC and the GuacaMol data set. We show that the GA is able to densely sample and explore the boundaries of the ZINC and the GuacaMol data set while staying within the property distributions of the reference data sets (see Figure S4). These new experiments show that the algorithm performs well under various objective functions that are motivated by real-world considerations.

---

### Official Review · AnonReviewer3 · 2019-10-31
**Official Blind Review #3**

**Rating:** 6

**Review:**

The authors present an algorithm to explore the generation of new organic molecules using a genetic algorithm.
The genetic algorithm will select molecules generated via mutation and cross over, according to a fitness function that includes a discriminator trained from generated molecules.
The mutation rule seems to be 50% insertions and 50% replacements of single SELFIES characters, plus a rule for the direct addition of phenyl groups with 4% chance.
No crossover rule is used.
The metric used is a maximum penalized logP score proposed by Gomez-Bombarelli, et al.
The work shows a huge improvement on this score using their proposed algorithm.
Additionally, the authors show some of the promise of the work in constrained optimization:
The algorithm can, with minor modifications, generate molecules with specific properties, and improve low scoring molecule.
There is also a study on the hyperparameter \beta - low beta tends generate high scoring molecules, and high beta generates molecules similar to the dataset.

Because of the novel genetic algorithm based search method, as well as the large improvement on prior literature, I am leaning towards an accept for this paper.
- Paper uses a metric well motivated by prior literature, J(m)
- There are constrained optimization studies showing different ways to use the algorithm
- The authors show the effect of the hyperparameter \beta on the algorithm - it seems to be difficult to interpolate between molecules in the dataset and molecules with high scores.
- All previous work have been with SMILES strings, where this work is the only one that builds up on SELFIES strings.
  It is not a sincere comparison - it would be better if the authors indicated which works use SMILES and which one uses SELFIES.
  A more sincere baseline would be to take a baseline model and apply it to SELFIES, while showing that the genetic algorithm based approach shows more viability.
  At least ORGAN seems to operate on SMILES strings, so it would be not too hard to change the underlying sequences to SELFIES strings.
- Some quantifiable metric of diversity of generated molecules might be a good analysis, it's unclear whether the model is simply memorizing a few high scoring molecules.


Two questions I have might improve the paper if they are answered in the text:
1) It seems to be the discriminator will decide whether molecules are in the dataset or not.
   If it is well trained and with a high penalty (e.g. time adaptive case), the GA will pick high J(m) molecules that are in the dataset.
   An analysis of the topline of the dataset might be useful - what is the J(m) of the best molecule in the dataset?
   If you simply seed a non-time adaptive GA with the best J(m) molecule, would it be able to reach the same levels?

2) Seeing the molecules generated in Figure 7d, it seems to imply to me that that algorithm is finding "bugs" in the simulation metric.
   Since the ICLR community does not have many chemists, it would be useful to make some claims about the chemical viability of these compounds outside of simulations.
   Are there molecules that are similar looking to the generated ones?
   Perhaps some analysis on whether the GA is overfitting to the synthetic metric of J(m) would be helpful.

Nits:
- Use uppercase ZINC in all cases to refer to the dataset
- Last sentence "An important future generalization..." awkward phrasing.


**Experience Assessment:**

I have read many papers in this area.

**Review Assessment: Checking Correctness Of Derivations And Theory:**

N/A

**Review Assessment: Checking Correctness Of Experiments:**

I assessed the sensibility of the experiments.

**Review Assessment: Thoroughness In Paper Reading:**

I read the paper at least twice and used my best judgement in assessing the paper.

---

> ### Author Response · Authors · 2019-11-15
> **Reply 1**
>
> Reviewer comment: The authors present an algorithm to explore the generation of new organic molecules using a genetic algorithm. The genetic algorithm will select molecules generated via mutation and cross over, according to a fitness function that includes a discriminator trained from generated molecules. The mutation rule seems to be 50% insertions and 50% replacements of single SELFIES characters, plus a rule for the direct addition of phenyl groups with 4% chance. No crossover rule is used. The metric used is a maximum penalized logP score proposed by Gomez-Bombarelli, et al. The work shows a huge improvement on this score using their proposed algorithm. Additionally, the authors show some of the promise of the work in constrained optimization: The algorithm can, with minor modifications, generate molecules with specific properties, and improve low scoring molecule. There is also a study on the hyperparameter \beta - low beta tends generate high scoring molecules, and high beta generates molecules similar to the dataset.
>
>
> Because of the novel genetic algorithm based search method, as well as the large improvement on prior literature, I am leaning towards an accept for this paper.
> - Paper uses a metric well motivated by prior literature, J(m)
> - There are constrained optimization studies showing different ways to use the algorithm
> - The authors show the effect of the hyperparameter \beta on the algorithm
> - it seems to be difficult to interpolate between molecules in the dataset and molecules with high scores.
> - All previous work have been with SMILES strings, where this work is the only one that builds up on SELFIES strings. It is not a sincere comparison
> - it would be better if the authors indicated which works use SMILES and which one uses SELFIES. A more sincere baseline would be to take a baseline model and apply it to SELFIES, while showing that the genetic algorithm based approach shows more viability. At least ORGAN seems to operate on SMILES strings, so it would be not too hard to change the underlying sequences to SELFIES strings.
>
> Answer: We thank the reviewer for his positive assessment of our work. We agree that one important difference between our work and previous models is the use of SELFIES replacing SMILES. The robustness of SELFIES allows us to operate using random mutations which decreases the dependence on hand-crafted or data set dependent mutation rules. We agree that a comparison with GANs or autoencoders trained on SELFIES instead of SMILES would be very valuable (VAEs are already shown in the SELFIES paper). We have been informed by the authors of the ORGAN paper that they work on integrating SELFIES in their models. Our paper mainly describes the idea of using SELFIES for exploration chemical space using genetic algorithms, rather than using VAE or GAN based models that are somewhat limited in their explorational capabilities.
>
> Reviewer comment: - Some quantifiable metric of diversity of generated molecules might be a good analysis, it's unclear whether the model is simply memorizing a few high scoring molecules.
>
> Answer:  This is an important comment. We have added an additional experiment, in which we have not only optimized for penalized logP, but also for drug-likeness score (QED) - Fig. 6. One clearly sees a wide distribution of the resulting molecules (green dots), which shows that the algorithm does not “simply memorize a few high scoring molecules” but generates a large variety of high performing molecules.

---

> ### Author Response · Authors · 2019-11-15
> **Reply 2**
>
> Reviewer comment: Two questions I have might improve the paper if they are answered in the text:
>
> 1) It seems to be the discriminator will decide whether molecules are in the dataset or not. If it is well trained and with a high penalty (e.g. time adaptive case), the GA will pick high J(m) molecules that are in the dataset. An analysis of the topline of the dataset might be useful - what is the J(m) of the best molecule in the dataset? If you simply seed a non-time adaptive GA with the best J(m) molecule, would it be able to reach the same levels?
>
> Answer: The discriminator is trained on (randomly drawn) molecules from an external data set (ZINC in our case) and molecules generated by the GA. The molecules from the reference data set are labeled with 1, the molecules generated by the GA are labeled with 0. The discriminator is trained for 10 epochs in each generation; the molecules from the reference data set change in each generation. Therefore, the discriminator constantly tries to learn how to differentiate between the molecules from the reference distribution and molecules generated by the GA, especially the high-scoring molecules that survive multiple generations. If seeded with the best J(m) molecule, the discriminator will quickly memorize this molecule (or similar molecules) and penalize them so that it will vanish from the population, forcing the GA to explore other regions of the chemical space. To clarify this to the reader, we added the following sentence to Section 3.3: Role of the discriminator:
> “The task of the discriminator thus is to memorize families of high performing molecules and penalize their fitness to force the GA to explore different regions in chemical space.”
>
> Reviewer comment: 2) Seeing the molecules generated in Figure 7d, it seems to imply to me that that algorithm is finding "bugs" in the simulation metric. Since the ICLR community does not have many chemists, it would be useful to make some claims about the chemical viability of these compounds outside of simulations. Are there molecules that are similar looking to the generated ones? Perhaps some analysis on whether the GA is overfitting to the synthetic metric of J(m) would be helpful.
>
> Answer: As mentioned in the reply to reviewer 1, the genetic algorithm is indeed exploiting the weaknesses of the logP and SA models that we use as objective function. We see this is an advantage rather than a bug as it shows that the algorithm is very quickly and efficiently exploring chemical space for solutions with high objective values. As discussed in Lehman et al. (https://arxiv.org/abs/1803.03453), strong explorative behaviour can lead to ML-inspired results that scientists would consider as surprising and creative.
>
> This will help the human researchers to improve their understanding of the optimization surface which will either lead to improvements in the definition of the objective function or to a better understanding of the underlying structure property relations. However, the GA is not used to fit the J(m) metric, but to recognize frequently occurring molecules and molecular motifs to penalize these and to enhance the explorative behaviour of the GA. We will add a discussion to this in our paper:
> “Visual inspection of the highest performing molecules shows that the GA is exploiting deficiencies of the (penalized) logP metric by generating chemically irrelevant motifs such as sulfur chains. While being of limited relevance for application, this nonetheless shows that the GA is very efficient in exploring the chemical space and finding solutions for a given task. Analysis of these solutions will help us to better understand and eventually improve objective functions. At the same time, surprising solutions found by an unbiased algorithm can lead to unexpected discovery or boost human creativity (Lehman et al. (2018)). While exploitative tasks that follow some reference database are significant for questions involving drug design, more explorative behaviour could be beneficial in domains such as solar cell design (Yan et al. (2018)), flow battery design (Yang et al. (2018)), and in particular for questions where datas sets are not available, such as design of targets in molecular beam interferometry (Fein et al. (2019)).”
>
> We furthermore added a new section to the paper with an objective function that mixes logP and QED. This new objective function is harder to be exploited, thus leading to more “realistic” looking molecules. The results and example molecules are shown in Section 4.5 and Figure 6.
>
> Reviewer comment: Nits:
> - Use uppercase ZINC in all cases to refer to the dataset
> - Last sentence "An important future generalization..." awkward phrasing.
>
> Answer: We changed the spelling to ZINC and adjusted the last sentence.

---

### Official Review · AnonReviewer2 · 2019-10-31
**Official Blind Review #2**

**Rating:** 8

**Review:**

The paper describes a model for the generation and optimization of molecules (de novo molecular design), which combines a genetic algorithm based on the SELFIES string representation and a trained discriminator.
The performance of the method is then shown on a small number of optimization tasks, and analyzed in detail.

Overall, this reviewer thinks this is a very interesting approach with a lot of potential, however, the experimental validation could be stronger.


Embedding in previous work:
Several key references should be added. In particular, this reviewer would suggest that the paper from the AZ group (https://arxiv.org/abs/1701.01329) is added,
 which first introduced the language model (LM)-based class of molecule generation models, and the RL paradigm of a molecule-generating agent, which takes actions in an environment to construct a molecule, receiving reward from an external scoring function.
Also, the previous paper by Liu et al (“Constrained Graph Variational Autoencoders for
Molecule Design”, NeurIPS 2018,  https://arxiv.org/pdf/1805.09076.pdf ) should be cited, in particular, because it highlights the competitiveness of the LM based baselines.

In Table 2, it would be great to include a reference to the DEFactor paper from Yoshua Bengio’s lab (https://arxiv.org/abs/1811.09766), which achieves comparable performance to the strongest baseline.

In addition, the authors should cite earlier work on using GAs for de novo design, which have a rich tradition in chemoinformatics since the 1990ies.

Experimental:

Re 4.1. In the unconstrained optimization task, several baselines are missing. In particular, a language model + RL baseline should be considered. This model has also shown strong performance (in terms of optimization and quality) in the state of the art benchmark set (Guacamol https://pubs.acs.org/doi/10.1021/acs.jcim.8b00839 ) -  the code is available on github

Can the authors explain why they believe the physchem only function ”log(P) - SA(m) - RingPenalty(m)” is a relevant scoring function for the tasks that are faced in real life drug and materials discovery? It would be great if the authors would consider using the Guacamol benchmarks which have been designed by domain experts to allow a more fine grained assessment on tasks relevant to practical drug and materials design, while still being tractable computationally. Also, the guacamol paper provides a benchmark to assess the quality of molecules, which is currently missing in the present work.

Re 4.2. And 4.3.
The long term experiment and the molecule class exploration analysis are great, and well presented. I like the analysis of how the different classes emerge. However, are you sure that the GA is not just exploited the deficiencies of the rdkit logP calculator (which itself is just a model) by creating the molecules with the long sulfur chains? Also, I am not sure that it is likely that one would be able to synthesize these molecules in the lab.

Re 4.4
The results on the constrained optimization are impressive, however, it is not really clear if the GA is not simply exploiting deficiencies in the scoring function again.


Overall, this reviewer believes the proposed model is very promising (in particular also because GA’s are cool), however, the experimental validation could be more rigorous, which can be addressed by considering the guacamol benchmarks. If this results can provided, this reviewer will raise their recommendation to 'accept' (regardless of outcome of the benchmark)

#####
After response:
Considering the authors' response, I adapt my recommendation to accept. I acknowledge the short time the authors have  to run additional experiments. nevertheless, I would suggest that the authors at least run more challenging benchmarks later and report the results e.g. as a blogpost or in a leaderboard.


**Experience Assessment:**

I have published in this field for several years.

**Review Assessment: Checking Correctness Of Derivations And Theory:**

I carefully checked the derivations and theory.

**Review Assessment: Checking Correctness Of Experiments:**

I carefully checked the experiments.

**Review Assessment: Thoroughness In Paper Reading:**

I read the paper thoroughly.

---

> ### Author Response · Authors · 2019-11-15
> **Reply 1**
>
> Reviewer comment: The paper describes a model for the generation and optimization of molecules (de novo molecular design), which combines a genetic algorithm based on the SELFIES string representation and a trained discriminator. The performance of the method is then shown on a small number of optimization tasks, and analyzed in detail.
>
> Overall, this reviewer thinks this is a very interesting approach with a lot of potential, however, the experimental validation could be stronger.
>
> Answer: We thank the reviewer for his positive assessment of the overall potential of our work. In the following, we will answer all questions and suggestions by the reviewer in detail.
>
> Reviewer comment: Embedding in previous work: Several key references should be added. In particular, this reviewer would suggest that the paper from the AZ group (https://arxiv.org/abs/1701.01329) is added, which first introduced the language model (LM)-based class of molecule generation models, and the RL paradigm of a molecule-generating agent, which takes actions in an environment to construct a molecule, receiving reward from an external scoring function. Also, the previous paper by Liu et al (“Constrained Graph Variational Autoencoders for Molecule Design”, NeurIPS 2018, https://arxiv.org/pdf/1805.09076.pdf ) should be cited, in particular, because it highlights the competitiveness of the LM based baselines.
>
> In Table 2, it would be great to include a reference to the DEFactor paper from Yoshua Bengio’s lab (https://arxiv.org/abs/1811.09766), which achieves comparable performance to the strongest baseline.
>
> In addition, the authors should cite earlier work on using GAs for de novo design, which have a rich tradition in chemoinformatics since the 1990ies.
>
> Answer: We added the additional references mentioned by the reviewer. In particular, we added references to Segler et al. and Liu et al. and we added the results on constrained optimization of the DEFactor model by Assouel et al. to Table 2.
>
> Addition to the paper: “Segler el al. first introduced molecule generating models based on language models and reinforcement learning, where actions in an environment are taken to construct a molecule, receiving reward from an external scoring function. This model has also shown strong performance in the GuacaMol benchmark.”
>
> Reviewer comment: Experimental:
>
> Re 4.1. In the unconstrained optimization task, several baselines are missing. In particular, a language model + RL baseline should be considered. This model has also shown strong performance (in terms of optimization and quality) in the state of the art benchmark set (Guacamol https://pubs.acs.org/doi/10.1021/acs.jcim.8b00839 ) - the code is available on github
>
> Answer: This is an interesting suggestion, and we cite the corresponding paper now. Thank you. The paper has not performed the task penalized logP task and therefore we have not added it to the list. However we clearly mention in the paper “This model has also shown strong performance in the GuacaMol benchmark.”
>
> Reviewer comment: Can the authors explain why they believe the physchem only function ”log(P) - SA(m) - RingPenalty(m)” is a relevant scoring function for the tasks that are faced in real life drug and materials discovery? It would be great if the authors would consider using the Guacamol benchmarks which have been designed by domain experts to allow a more fine grained assessment on tasks relevant to practical drug and materials design, while still being tractable computationally. Also, the guacamol paper provides a benchmark to assess the quality of molecules, which is currently missing in the present work.
>
> Answer: This is a very important comment to which we agree. The penalized logP task, which became a widely used objective in the literature recently, does not take into account properties that are significant for some important optimization task in the area of pharma chemistry. We are thankful for the suggestion of the GuacaMol benchmarks, which consists of many similarity- and rediscovery-measures, that are significant tasks in the application of drug discovery and optimization. While the limited amount of time makes it impossible for us to reproduce the full set of benchmarks suggested in the GuacaMol paper, we show important comparisons of our results in comparison with GuacaMol data sets and add benchmarks to our work that are also relevant for drug discovery (as requested by Reviewer 1). Specifically, we performed a new experiment to the paper that shows the simultaneous optimization of the penalized logP score and the drug-likeness score (QED). The results of this experiment are shown in Figure 7. The experiment demonstrates that we can simultaneously optimize logP and QED, and therefore densely sample and explore the boundaries of the ZINC and the GuacaMol data set.

---

> > ### Comment · AnonReviewer2 · 2019-11-15
> > **reply**
> >
> > Thank you very much for the response, and edits to the paper.
> >
> > While I still believe the experiments could be extended (and as a community, we need to move to more challenging benchmarks at these tasks, but that is another conversation to be had), I acknowledge the short rebuttal times.
> >
> > I've changed my recommendation accordingly.

---

> ### Author Response · Authors · 2019-11-15
> **Reply 2**
>
> Reviewer comment: Re 4.2. And 4.3. The long term experiment and the molecule class exploration analysis are great, and well presented. I like the analysis of how the different classes emerge. However, are you sure that the GA is not just exploited the deficiencies of the rdkit logP calculator (which itself is just a model) by creating the molecules with the long sulfur chains? Also, I am not sure that it is likely that one would be able to synthesize these molecules in the lab.
>
> Answer: We agree with the reviewer that our genetic algorithm quickly found ways to exploit the logP and the SA models. However, we see this as a strength, rather than a bug. As discussed in Lehman et al. (https://arxiv.org/abs/1803.03453), it is very valuable for human researchers to have methods that can quickly show the limits and extreme cases of optimization tasks to learn more about the underlying optimization surface and to develop better hypotheses and models for designing optimal solutions in a rational way. As argued there, strong explorative behaviour can lead to ML-inspired results that scientists would consider as surprising and creative.
> We agree that the sulfur chains generated by the GA are not of practical interest and challenging to synthesize in the lab. However, this points at a deficiency of the SA scoring function that we used, rather than being a weakness of our algorithm. For that reason, as mentioned above, we made the additional experiment involving the much better developed QED score, suggested in GuacaMol.
>
> Reviewer comment: Re 4.4 The results on the constrained optimization are impressive, however, it is not really clear if the GA is not simply exploiting deficiencies in the scoring function again.
>
> Answer: We again agree that the model might exploit deficiencies of the scoring function. To compare with literature results we use the same (deficient) fitness function and thus see it as an advantage that our model quickly learns to fully use (or exploit) the objective function. In addition, by adding the new comparison with the GuacaMol data set based on the QED property, we are confident that the model can perform well on a multitude of different tasks, going well beyond the simple penalized logP function toward real-world problem in drug and materials design.
>
> Reviewer comment: Overall, this reviewer believes the proposed model is very promising (in particular also because GA’s are cool), however, the experimental validation could be more rigorous, which can be addressed by considering the guacamol benchmarks.
>
> Answer: We thank the reviewer again for the very constructive suggestions. We hope that our changes to the manuscript can convince the reviewer that our model can also be applied to objective functions which are of interest in real-world applications.

---

### Decision · Program_Chairs · 2019-12-19

**Decision:**

Accept (Poster)

**Comment:**

Paper received reviews of A, WA, WR. AC has carefully read all reviews/responses. R1 is less experienced in this area. AC sides with R2,R3 and feels paper should be accepted. Interesting topic and interesting problem. Authors are encouraged to strengthen experiments in final version.